# THALAMUS: A BRAIN-INSPIRED ALGORITHM FOR BIOLOGICALLY-PLAUSIBLE CONTINUAL LEARNING AND DISENTANGLED REPRESENTATIONS

**Ali Hummos**
Brain and Cognitive Sciences Department
Massachusetts Institute of Technology
Cambridge, MA 02139
`ahummos@MIT.edu`

## ABSTRACT

Animals thrive in a constantly changing environment and leverage the temporal structure to learn well-factorized causal representations. In contrast, traditional neural networks suffer from forgetting in changing environments and many methods have been proposed to limit forgetting with different trade-offs. Inspired by the brain thalamocortical circuit, we introduce a simple algorithm that uses optimization at inference time to generate internal representations of the current task dynamically. The algorithm alternates between updating the model weights and a latent task embedding, allowing the agent to parse the stream of temporal experience into discrete events and organize learning about them. On a continual learning benchmark, it achieves competitive end average accuracy by mitigating forgetting, but importantly, by requiring the model to adapt through latent updates, it organizes knowledge into flexible structures with a cognitive interface to control them. Tasks later in the sequence can be solved through knowledge transfer as they become reachable within the well-factorized latent space. The algorithm meets many of the desiderata of an ideal continually learning agent in open-ended environments, and its simplicity suggests fundamental computations in circuits with abundant feedback control loops such as the thalamocortical circuits in the brain

## 1 INTRODUCTION

Animals thrive in a constantly changing environmental demands at many time scales. Biological brains seem capable of using these changes advantageously and leverage the temporal structure to learn causal and well-factorized representations (Collins & Koechlin, 2012; Yu et al., 2021; Herce Castañón et al., 2021). In contrast, traditional neural networks suffer in such settings with sequential experience and display prominent interference between old and new learning limiting most training paradigms to using shuffled data (McCloskey & Cohen, 1989). Many recent methods advanced the flexibility of neural networks (for recent reviews, see Parisi et al. (2019); Hadsell et al. (2020); Veniat et al. (2021)). However, in addition to mitigating forgetting, several desirable properties in a continually learning agent have been recently suggested (Hadsell et al., 2020; Veniat et al., 2021) including: **accuracy** on many tasks at the end of a learning episode or at least **fast adaptation and recovery** of accuracy with minimal additional training. The ideal agent would also display **knowledge transfer** forward, to future tasks and backwards to previously learned tasks, but also transfer to tasks with slightly different computation and or slightly different input or output distributions (Veniat et al., 2021). The algorithm should **scale** favorably with the number of tasks and maintain **plasticity**, or the capacity for further learning, Finally, the agent should ideally able to function **unsupervised** and not rely on access to task labels and task boundaries (Hadsell et al., 2020; Rao et al., 2019). We argue for another critical feature: **contextual behavioral**, where the same inputs may require different responses at different times, a feature that might constrain the solution space to be of more relevance to brain function and to the full complexity of the world. A learning agent might struggle to identify reliable contextual signals in high dimensional input space,

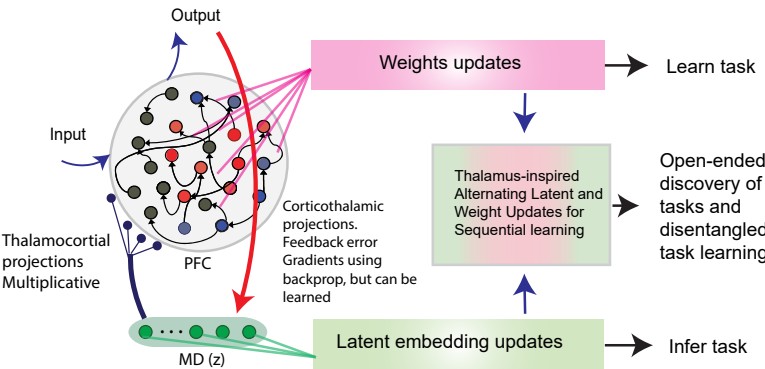

Figure 1: Model schematic. Controlled triggering of latent updates and weight updates and alternating them allows for unsupervised task discovery. Latent updates retrieve previously learned tasks or choose a new embedding for new ones (PFC: prefrontal cortex, MD: mediodorsal thalamus)

if they are knowable at all, and with many contextual modifiers it might not be feasible to experience all combinations sufficiently to develop associative responses.

Neuroscience experiments have revealed a thalamic role in cognitive flexibility and switching between behavioral policies (Schmitt et al., 2017; Mukherjee et al., 2021). The prefrontal cortex (PFC), linked to advanced cognition, shows representations of task variables and input to output transformations (Johnston et al., 2007; Mansouri et al., 2006; Rougier et al., 2005; Rikhye et al., 2018), while the medio-dorsal thalamus shows representations of the task being performed (Schmitt et al., 2017; Rikhye et al., 2018), and uncertainty about the current task (Mukherjee et al., 2021). The mediodorasal thalamus, devoid of recurrent excitatory connections, and with extensive reciprocal connections to PFC is thought to gate its computations by selecting task relevant representations to enable flexible behavioral switching (Wang & Halassa, 2021; Hummos et al., 2022). The connections from cortex to thalamus engage in error feedback control to contextualize perceptual attention (Briggs, 2020), motor planning (Kao et al., 2021), and cognitive control (Halassa & Kastner, 2017; Wang & Halassa, 2021) and representations of error feedback can be observed in the thalamus (Ide & Li, 2011; Jakob et al., 2021; Wang et al., 2020). Moreover, these thalamic representations can be composed to produce complex behaviors (Logiaco et al., 2021).

In this paper, we take inspiration from the thalamocortical circuit and develop a simple algorithm that uses optimization at inference time to produce internally generated contextual signals allowing the agent to parse its temporal experience into discrete events and organize learning about them (Fig 1). Our contributions are as follows. We show that a network trained on tasks sequentially using the traditional *weight updates*, with task identifiers provided, can be used to identify tasks dynamically by taking gradient steps in the latent space (*latent updates*). We then consider unlabeled tasks and simply alternate weight updates and latent updates to arrive at Thalamus, an algorithm capable of parsing sequential experience into events (tasks) and contextualizing its response through the simple dynamics of gradient descent. The algorithm shows generalization to novel tasks and can discover temporal events at any arbitrary time-scale and does not require a pre-specified number of events or clusters. Additionally, it does not require distinction between a training phase or a testing phase and is accordingly suitable for open-ended learning.

## 2 MODEL

We consider the setting where tasks arrive sequentially and each task k is described by a dataset $D^k$ of inputs, outputs, and task identifiers $(x^k, y^k, i^k)$. We examine both settings where the task identifier is and is not available to the learning agent as input.

As a learning agent, we take a function $f_\theta$ with parameters $\theta$ that takes input $x^k$ and latent embedding $\mathbf{z}$:

$$\hat{\mathbf{y}}^k = f_\theta(\mathbf{x}^k, \mathbf{z}) \tag{1}$$

We define some loss $\mathcal{L}$ as a function of predicted output $\hat{\mathbf{y}}$ and ground truth $\mathbf{y}$:

$$\mathcal{L}(\hat{\mathbf{y}}, \mathbf{y}) \tag{2}$$

To optimize the model we define two updates of interest, updating the model parameters $\theta$ in the traditional sense according to their gradient $\Delta\theta$ (*weight update*), and updating the latent embedding $\mathbf{z}$ along the direction of the the error gradients $\Delta\mathbf{z}$ (*latent update*), by backpropagating gradients from $\mathcal{L}$ through $f_\theta$, but keeping $\theta$ unchanged.

$$\Delta\theta \sim \frac{d\mathcal{L}}{d\theta}, \qquad\qquad \Delta\mathbf{z} \sim \frac{d\mathcal{L}}{d\mathbf{z}} \tag{3}$$

We showcase our algorithm by parametrizing $f_\theta$ with a vanilla RNN model, as an abstraction of PFC (although note applicability to any neural network architecture). Pre-activation of the RNN neurons $v(t)$ evolve in time as follows (bias terms excluded for clarity):

$$\tau_v \frac{\mathrm{d}}{\mathrm{d}t}\mathbf{v}(t) = -\mathbf{v}(t) + W^{in}\mathbf{x}(t) + W^r\phi(\mathbf{v}(t)) \odot W^z\mathbf{z} \tag{4}$$

Where v is the pre-nonlinearity activations, $\phi$ is a relu activation function, $W^r$ is the recurrent weight matrix, $\mathbf{x}(t)$ is the input to the network at time t, $W^{in}$ are the input weights, $\mathbf{z}$ is the latent embedding vector (representing the thalamus), and $W^z$ are the weights from the latent embedding initialized from a Bernoulli distribution $w_{i,j} \sim B(p = 0.5)$ (or a rectified Gaussian). These projections had a multiplicative effect, consistent with amplifying thalamocortical projections (Schmitt et al., 2017). Network output $\hat{\mathbf{y}}$ was a projection layer from the RNN neurons

$$\hat{\mathbf{y}} = \phi(W^{out}\phi(\mathbf{v}(t))) \tag{5}$$

When task identifiers $i^k$ were available they were input directly to the latent embedding $\mathbf{z}$ ($\mathbf{z} = i^k$) vector as one-hot encoded vectors. When task identifiers were not available, $\mathbf{z}$ was set to a uniform value. The mean squared error loss was used for both latent and weight updates.

**Biological-plausibility.** While we rely on backpropagation to obtain gradients with respect to the latent embedding $\mathbf{z}$, these gradients can be learned using a second neural network (Marino et al., 2018; Greff et al., 2019), which would compare the output to feedback from environment and produce credit assignment signals to the latent embedding. The model would be implemented as two neural networks. One does the task computations and is gated by a latent embedding, and a second that compares the output to feedback from environment and produces gradients for the latent embedding. These learned credit assignment projections then would correspond to the corticothalamic projections while the weights from the latent embedding to the RNN would correspond to the thalamocortical projections. Once the credit assigning corticothalamic projects are learned, the structure can now show ability to adapt its behavior with no need for backpropagation or any parameters updates. The circuit now adapts by updating the latent embedding, rather than re-learning parameters, which matches modern formulations of adaptive behavior in neuroscience (Heald et al., 2021). We assume learning relies on the credit assignment mechanisms in the brain, though these are yet to be fully understood (Lillicrap et al., 2020). For this work, we use backpropagation to maintain generality, as learning the credit assignment network might impart a level of domain specialization.

## 3    EXPERIMENTS

We first describe continual learning dynamics in an RNN model trained with usual *weight update* paradigm, then we demonstrate the flexibility of taking gradient descent steps in the latent space (*latent update*) in the resulting pretrained model. Second, we consider unlabeled tasks stream and combine both updates in a simple algorithm capable of meeting most desiderata of an ideal continual

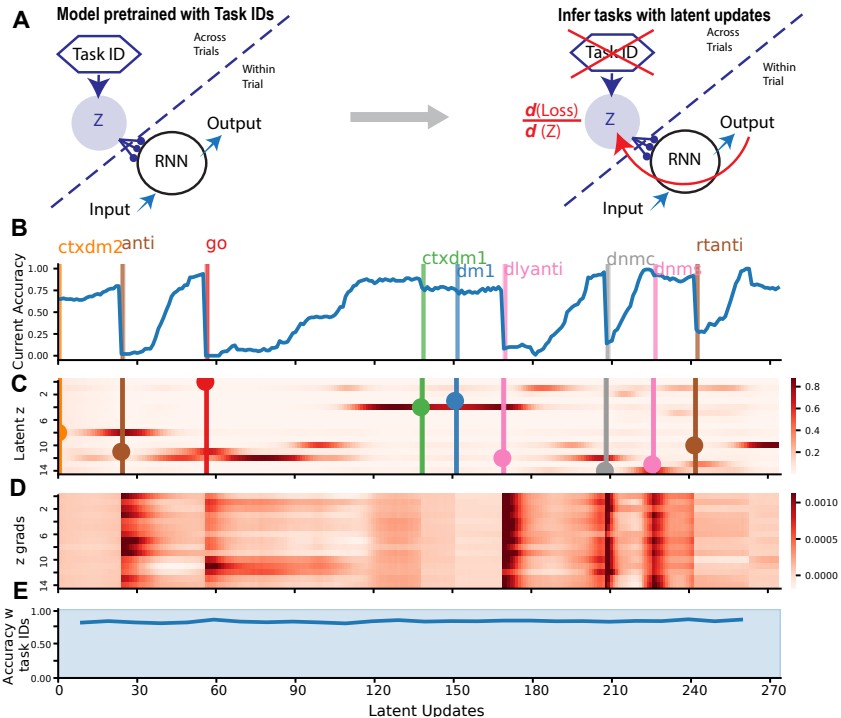

Figure 2: (A) Schematic of the experiment, model pretrained with task IDs and transitioning to identifying tasks through gradients backpropagated from the loss function to update the task embedding z (latent updates). (B) Current accuracy as the latent is updated repeatedly using only one batch from the current task. We switch to the next task once accuracy criterion is reached. (C) current values of the latent task embedding vector z. Circular markers indicate the one-hot task ID used during pretraining. (D) Gradients of loss with respect to z. (E) Average accuracy on all tasks if we were to use the original task IDs used during pretraining, shows no forgetting of previous tasks.

learning agent, including learning disentangled representations that factorize along the structure of the environment (Fig 1). We test the model on a set of cognitive neuroscience tasks, and the standardized split MNIST dataset.

The algorithm was implemented in PyTorch and optimized with the Adam optimizer (Kingma & Ba, 2017) (learning rate 0.001 for weights updates and latent updates). Simulations were run at the MIT OpenMind computing cluster and we estimate in the order of $10^4$ GPU hours were used in developing the model. Code is available at GitHub.com/hummosa/Thalamus.

### 3.1 PRETRAINING RNN WITH TASK IDENTIFIERS, REHEARSAL, AND TRAINING TO CRITERION

We start by pretraining our vanilla RNN with task identifier input on a collection of 15 cognitive tasks commonly used in neuroscience and cognitive science experiments (Yang et al., 2019), although we also present results on split MNIST dataset in a later section. The tasks are composed of trials with varying sequence lengths and input with brief noisy stimuli representing angles. The input-output computations require cognitive abilities such as working memory, perceptual decision making, contextual decision making amongst others, and importantly the same input may require opposite outputs depending on the task (example tasks shown in Fig A17). We train tasks only to criterion to prevent over-training and use rehearsal, a ubiquitous feature in animal learning, as they acquire expertise in new domains. The network achieves high end average accuracy on all tasks successfully (details of pretraining in Appendix A.2, Figs. A7, A8). This RNN trained with one-hot encoded task identifiers became proficient at all 15 tasks at the end of training, and we refer to it as the *pretrained network*.

### 3.2 PRETRAINED NETWORK CAN INFER TASKS THROUGH LATENT UPDATES

To handle unlabeled tasks, one can possibly transition this pretrained network through weight updates and due to its accelerated ability at learning new tasks and rehearsing previous tasks (Fig A7), it can adapt its weights rapidly to transition between tasks with a small number of batches (Fig A9).

This is analogous to fine-tuning pretrained vision and language models to new data distributions. However, this assumes there is enough data to describe the new distribution fully (which is the case in our simple dataset), and the network would have to constantly update weights as it transitions between distributions, making it prone to forgetting. Most critically, the network is not inferring a change in distribution and has no representation of it, but merely adapting the weights to it.

Here, we consider an alternative solution using latent updates to adapt to changing data distribution. By consuming only one batch, we take gradient steps in the latent space until accuracy reaches criterion again, and then allow the model to process incoming batches of the new task. Thereby the switch consumes only one batch (although other datasets and environments might require more data to accurately disambiguate context). We note several interesting aspects of this algorithm:

☐ The algorithm correctly switches the RNN to solve incoming tasks without forgetting any of its previous learning as evident by no decrease on its accuracy to perform all 15 tasks if given the task identifiers $i^k$ again (Fig 2E).

☐ At moments displays serial search behavior through several possible task labels before converging on the correct one, as a by-product of Hamiltonian dynamics (Fig 2C, e.g. in the solution of cxtdm1 from latent update 60 to 130).

The algorithm might not solve the tasks using the embedding vectors it was pretrained on but rather using any latent vectors that perform well given its recent trajectory in the latent space (Fig 2C). Interestingly, even though it was pretrained on uninformative latent space (equally distributed one-hot vectors), the latent updates retrieve a rich latent space that is functionally organized and clusters tasks according to similarity and compatibility (Fig 3).

Pretraining the network however assumes one has access to data where data distribution changes are labelled across the correct dimensions. With this motivation, we next describe an algorithm that drops the requirement for any task labels.

### 3.3 ALTERNATING WEIGHT AND LATENT UPDATES TO DISCOVER TASK REPRESENTATIONS IN A STREAM OF UNLABELED TASKS

We combine weight updates and latent updates in a simple algorithm that can recover the latent variables in the environment as it follows the dynamics of the data distribution in time. We start with a naive RNN attempting to solve a stream of incoming unlabeled tasks. This network was freshly initialized and had no previous training on any tasks. It learns the first task through weight updates, and as the average accuracy begins to increase, any sudden drops from running average by more than a specified amount (0.1) would trigger switching to latent updates. We take a specified number of gradient descent steps in the latent space. If average accuracy has not recovered, the algorithm then goes back to learning through weight updates (see Algorithm A1). More intuitively, once the current latent embedding no longer solves the current task, use latent updates to reach at another one. If the task was seen before likely this will solve the current task and you move on. If not, then it is likely a new task and gradient de-

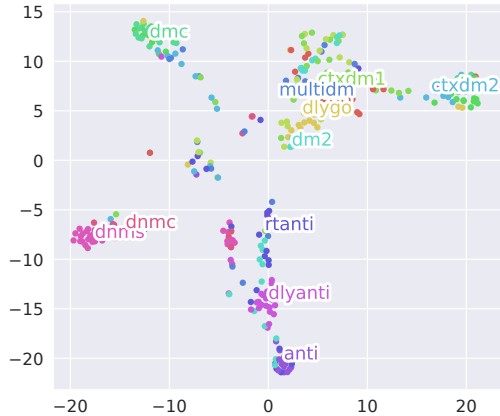

Figure 3: Dimensionality reduction with TSNE of the latent vectors after latent updates on each task. Data point colored by the ground truth task ID

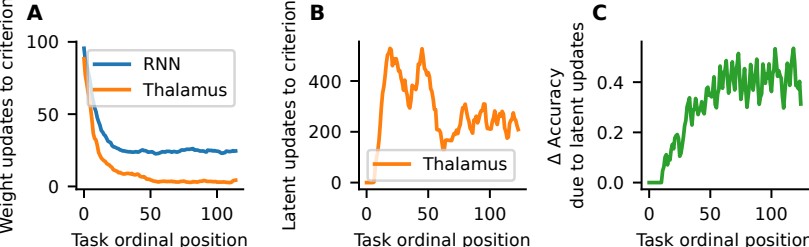

Figure 4: Thalamus performance on an unlabeled sequence of tasks. A) Comparing number of weight updates needed to solve tasks along the training sequence for Thalamus and baseline RNN. Thalamus reliance on weight updates begins to vanish. B) Latent updates needed for each task in the sequence. C) The change in accuracy after the latent update loop. Latent updates contribute increasingly more to recovering accuracy at task transitions. Results averaged over 20 random seeds.

scent arrived at an untrained embedding. The algorithm then goes back to weight updates and learns the task with the loaded embedding.

These dynamics allow the algorithm to successfully uncover the latent structure of the environment (discrete stream of tasks) without supervision, while remaining invariant to the time scale and not requiring a predefined number of clusters or events. Successful convergence leads the algorithm to require fewer weight updates and eventually, handle the stream of unlabelled tasks through latent updates only (Fig 4, example run in Fig A10).

For quantitative comparison with other models, we refer the reader to the next section for experiments on split MNIST. For the cognitive task, we noted that continual learning models were not designed to handle contextual behavior. For example, methods for task-agnostic continual learning (Rao et al., 2019; Zeno et al., 2021; Achille et al., 2018) have a task inference network that maps inputs to task identity, which means that transitioning between tasks requiring different output for the same inputs would require training and weight updates. To quantify results on cognitive tasks, we compared Thalamus to an RNN that used weight updates to transition between tasks, as a proxy. We run both Thalamus and the RNN on a stream of unlabeled tasks drawn from 5 cognitive tasks. Thalamus reliance on weight updates to transition between tasks drops rather rapidly after the first few times it sees a task, while reliance on latent updates begins to dominate (Fig 4). While Thalamus required more model runs due to the latent updates, it had higher average accuracy due to recovering promptly from dips in accuracy at task transitions (Table 1). More importantly, we evaluated on the unseen 10 cognitive tasks, with the weights frozen, and Thalamus through latent updates only was able to solve 27% of unseen tasks (see quantification on a more standardized benchmark next section) (Table 1).

A vexing finding in brains is that despite internal consistency and intact function, recordings show that, to an outside observer, brain representations seem to drift and change over days and weeks (for e.g. see Marks & Goard (2021)), causing difficulties, for example, in designing a brain controller interface (BCI) (Mridha et al., 2021). By training a classifier to predict task identifier from latent z vector, we show that the internal-generated representations in our model reveal similar dynamics (Fig 5).

## 3.4 COMPARISON ON EXTERNAL BENCHMARK: SPLIT MNIST

We compare our model to other continual learning methods on a popular continual learning benchmark that splits MNIST digits into 5 pairs, making each pair into a binary classification task (0 or 1, chosen arbitrarily) (LeCun & Cortes, 2010), (referred to as 'domain incremental learning' Ven & Tolias (2019)). Models are shown an image at a time from each task and are required to classify it as 0 or 1. Importantly, models have no access to task identifiers and are trained on the 5 tasks sequentially. As Thalamus embodies a slightly different paradigm meant to operate in dynamic environments, testing its performance on a task requires prompting with a batch from the training-set of the same task, so it can adapt its latent embedding z, while keeping the weights frozen (i.e.

Table 1: Model comparison on NeuroGym tasks: Thalamus vs. RNN, from the last 20 blocks in a 5-task sequence learning over 20 random seeds. Solved defined as > 0.8 accuracy

| Model | Weight updates | Avg accuracy | Total updates (compute) | Ratio novel tasks solved |
|-------|----------------|--------------|-------------------------|--------------------------|
| RNN | $23.73 \pm 1.21$ | $0.91 \pm 0.00$ | $23.73 \pm 1.21$ | $0.21 \pm 0.12$ |
| Thalamus | $3.64 \pm 0.50$ | $0.96 \pm 0.00$ | $271.56 \pm 55.28$ | $0.27 \pm 0.14$ |

Table 2: End average accuracy (tested on all 5 tasks, mean ($\pm$ SEM)) on the split MNIST binary classification tasks. Performance results from Ven & Tolias (2019)

| Method | Accuracy | Prompt with a training batch |
|--------|----------|------------------------------|
| Upper bound (iid) | 98.42 ($\pm$ 0.06) | No |
| EWC | 63.95 ($\pm$ 1.90) | No |
| SI | 65.36 ($\pm$ 1.57) | No |
| LwF | 71.50 ($\pm$ 1.63) | No |
| DGR | 95.72 ($\pm$ 0.25) | No |
| RtF | 97.31 ($\pm$ 0.11) | No |
| Thalamus (Ours) | 97.71 ($\pm$ 0.13) | Yes |

it requires feedback from the environment whether as a supervised signal or a reinforcement reward). Thalamus shows competitive end average accuracy (Table 2) when compared to the popular regularization-based methods synaptic intelligence (SI) (Zenke et al., 2017), elastic weight consolidation (EWC) (Kirkpatrick et al., 2017) although these methods are at a great disadvantage here as they generally require access to task labels and task boundaries. We also compare to three generative rehearsal methods that are of relevance to neuroscience theories: deep generative replay (DGR) (Shin et al., 2017), learning without forgetting (LwF) (Li & Hoiem, 2018), and the model with the current state of the art, replay through feedback (RtF) (Ven & Tolias, 2019) (Table 2). Of note, other more recent works do not report results on MNIST and moved on to more complex datasets requiring specific optimizers, data augmentations, and loss functions, making comparison to their core architecture difficult (Pham et al., 2021; He & Zhu, 2021), and as we argue here, not all desiderata of continual learning have been met for the simpler MNIST-based tasks. Further experimental details can be found in Appendix A1.

## 3.5 KNOWLEDGE TRANSFER AND CONTEXTUAL BEHAVIOR

To assess knowledge transfer we use the split MNIST task but limit the model to learning only the first 4 tasks and then test on the last unseen task. The model requires one batch from the training set of the last task to update its latent z (while weights frozen), and achieves a 91.14±3.73 accuracy on the last task (avg accuracy on all tasks 93.62 ($\pm$ 0.30)) (example run in Fig A11). Through learning the first four tasks the MLP and the latent z become a function parametrizable along relevant dimensions to solve related tasks, while the weights are frozen. To highlight this difference compared to other methods we plot the accuracy on task 5 as models learn the tasks sequentially (Fig 6). This generalization relies on balance between adapting via weight updates or latent updates, and conflict between the weight updates required for each task leads to richer semantics offloaded to late updates. To showcase these effects, we consider that continual learning literature recommends using stochastic gradient descent (SGD) with momentum and a low learning rate to limit interference in weight updates thereby limiting forgetting of earlier tasks (Hsu et al., 2019; Mirzadeh et al., 2020). Thalamus shows the opposite effect, overall accuracy drops (to 83%) but we see a disproportionately higher drop in accuracy of the unseen fifth task (drops to 67%) when using SGD with $10^{-4}$ learning rate.

**Contextual split MNIST:** Finally, we introduce a new challenge by taking the first 3 tasks from split MNIST but then adding task 4 and 5 by using images from tasks 1 and 2 but flipping the labels. It is trivial to show that other continual learning methods, which rely on input images to infer context, achieve accuracy around 59%, but Thalamus maintains similar accuracy (96.09 ±0.83).

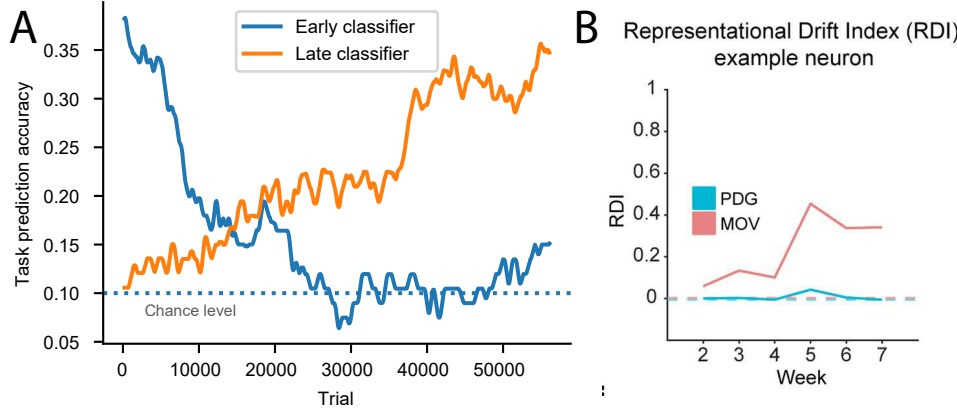

Figure 5: (A) Representational drift in the network trained on 10 tasks, measured by training a logistic regression classifier to predict task identity from the latent z embeddings, either from early (first 1000 batches) or late batches and tested accuracy on all. (B) Drift measured in the responses of a visual cortex neuron in response to natural movie (MOV) or passive drifting gratings (PDG). Adapted from Marks & Goard (2021)

## 4  RELATED WORK

Our work adds to a growing continual learning literature, we refer the reader to recent reviews (Hadsell et al., 2020; Parisi et al., 2019), and only discuss papers of specific relevance to our algorithm. Continual learning of unlabeled tasks: several methods have been proposed for continual learning when task boundaries and task identities are not known and most methods use a variational inference approach (Rao et al., 2019; Zeno et al., 2021; Achille et al., 2018) to infer tasks and map current data points to a categorical variable representing task identity. Most methods require learning a task inference network to a latent space, running inference on the latent space and decoding to output. To handle new tasks, they keep a buffer of unexplained data points and instantiate a new categorical task once the buffer fills. In contrast, Thalamus relies on simply training a network to solve a given task, and task inference is done by taking gradients through its computations. The shape of the latent space inferred by latent updates is fully specified by the forward network weights. Additionally, by using a continuous vector to represent tasks, it can handle

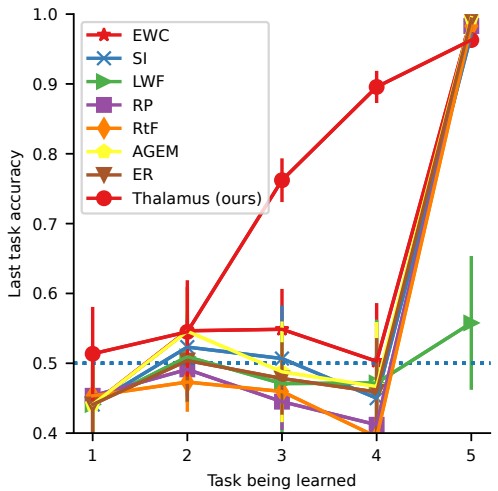

Figure 6: Measuring accuracy of the last task only as various models learned the sequence of tasks from 1 to 5. Simulations run as specified in Ven & Tolias (2019).

an open ended number of tasks naturally. A critical difference is that these methods are unable to handle contextual behavior which maps the same input to different outputs in different contexts.

Thalamus is intimately related to the theory of predictive coding in neuroscience (Rao & Ballard, 1999) and our gradient descent updates to the thalamus vector are conceptually similar to propagation of prediction errors. The algorithm also connects to adaptive learning from behavioral science (Yu et al., 2021), and to perspectives in neuroscience that consider learning as interaction between learning in the 'weight space' and 'activity space' (Sohn et al., 2021; Heald et al., 2021). Our work

instantiates these concepts by learning by using gradient descent for latent updates. Similar use of gradient descent to update latent variables was proposed as model for schizophrenia by Yamashita & Tani (2012) and more recently to retrieve language instructions for cognitive tasks (Riveland & Pouget, 2022). Our work uses these methods as a component in a continually learning agent to learn disentangled representations.

The concept of fast weights also prescribes changes in the model parameters at inference time to adapt computations to recent context (Ba et al., 2016), however our model extends this to fast changes in the neural space and uses credit assignment and gradient descent at inference time. $RL^2$ algorithm uses meta-learning to instantiate an RNN that can eventually adapt itself to a changing environment (Duan et al., 2016). In common with Thalamus, it also instantiates a learning system that operates in neural activity space to adapt its computations to ongoing task demands. Hypernetworks utilize one network that generates parameters of another network to similarly adapt its computations (Ha et al., 2016). Both algorithms use gradient flow through the task loss to determine the optimal neural activity, however they use gradients heavily during the extensive training of the meta-learner (hypernetwork) and use only forward propagation during inference. Our algorithm combines use of gradient updates with forward propagation at inference time resulting in a much simpler algorithm and continued adaptability with no specified training phase. Deep feedback control operates on similar concepts by calculating an inverse of a neural network computation and allowing a deep feedback controller to dynamically guide the network at inference time (Meulemans et al., 2022). Our model uses back-propagation through the neural network and operates on gradients instead of estimating a function inverse. Finally, the algorithm is also related to energy-based models that also rely on optimization-based inference (Mordatch, 2018) but our algorithm does not require the lengthy training with negative samples to train the energy-based model.

## 5 CONCLUSIONS

Thalamus is a relatively simple algorithm that can label ongoing temporal experience and then adapt to future similar events by retrieving those internally generated labels. A notable limitation is since we use a simple latent space, the form of knowledge transfer between tasks is quite limited conceptually to adapting the same computations if they apply to a new or old task. A richer form of knowledge transfer would emerge with a latent space that allows for compositionality over computational primitives. Future work will consider latent spaces with hierarchical or recurrent structures. Another possible extension is once the gradients of the latent embedding are approximated with a second neural network, the algorithm can be deployed in hardware, allowing it to show adaptive behavior and 'learn' without requiring backpropagation or weight updates. The algorithm shows several conceptually interesting properties, and the representations it might learn from more complex and naturalistic datasets will be of interest to compare to brain representations.

### ACKNOWLEDGMENTS

Thanks to Matthew Nassar, Brabeeba Wang, Ingmar Kanitscheider, Robert Yang, and Michael Halassa for insightful discussions and valuable comments on earlier drafts.

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

# A    Appendix

## A.1    Training details and hyperparamters for Thalamus training with no task identifiers

For the cognitive tasks neurogym dataset, we used a 356-dimensional RNN and a dt of 100ms and tau of 200ms. Recurrent weight matrix initialized as an identity matrix (Le et al., 2015; Yang et al., 2019). Multiplicative binary projections from z to the RNN had sparsity of 0.5. The training sequence used 5 randomly sampled cognitive neurogym task. We initially present the tasks to be learned in the same order twice for 200 batches each and allow the model to adapt with weight updates only. The model then sees the same sequence repeatedly. The first 3 repetitions are with a block size of 200 and then we lower it to 100 for efficiency and easier visualization.

For the split MNIST dataset and comparison with other models, we employ the same architecture and hyperparameters reported in Ven & Tolias (2019); Hsu et al. (2019). The classifer network had two fully-connected layers with 400 ReLU units and a 2-unit output layer. We used a 10 dimensional latent z embedding space that projected multiplicative binary gates with 0.8 sparsity. The same gating weights were used for both layers for simplicity. For this dataset we follow a more incremental sequencing of the tasks to align better with other methods on the same dataset. Tasks are incrementally added and the tasks seen so far are rehearsed twice after each new task. Only latent updates were used when evaluating on unseen tasks. See Fig A10 for an example run.

## A.2    Pretraining the network for sections 3.1 and 3.2 with task identifiers

We provide an example learning session of a sequence of 15 tasks with rehearsal and training to criterion (Fig A8). During training, accuracy on all other tasks was tested every 10 batches using the task IDs as input to indicate the testing task. Ablations show that task identifiers, training to criterion and rehearsal are the minimal components for this network and this dataset. As the network learns more tasks we see a decrease in the batches needed to acquire new tasks or rehearse early ones (7), consistent with previous literature (Goudar et al., 2021; Davidson & Mozer, 2020).

## A.3    Rapid adaptation of pretrained network

We provide an example of training the network using weight updates. After pretraining with task IDs it can now transition its behavior without task IDs, by adapting its parameters rapidly to handle oncoming tasks and requiring few updates to achieve criterion performance on each task (Fig A9). But see main text for discussion of the limitations of this approach to adaptable behavior.

## A.4    Thalamus algorithm

For completeness we provide the algorithm details below in Algorithm 1.

Clustering the latent embedding values Thalamus assigned to tasks shown a somewhat difficult to interpret distribution, with tasks having two embedding clusters, sometimes shared with another task. Some functional organization along task computations but also tasks seem to have limited overlap (Fig A12). Further study of the topography of the latent surface under different conditions and creating more compositional structures will be the subject of future studies.

To clarify the need for the latent updates to be triggered judiciously and only when an environment change is suspected, we lesion the algorithm by removing the accuracy criterion and simply updating the weights and then the latents every trial (Fig A13). The two updates seem to disrupt each other, with weight updates need to transition between tasks never approaching zero (Fig A13A), and on average, the latent updates seem to decrease accuracy (Fig A13C).

## A.5    Further examination of the latent embedding to RNN weights $\mathbf{W}^z$

For the results reported in the paper we draw $\mathbf{W}^z$ from a Bernoulli distribution with p=0.5 for neurogym tasks and 0.2 for split MNIST. To examine relaxing such a requirement we ran Thalamus on a sequence of 5 neurogym tasks using a Wz drawn from a Gaussian distribution with a mean of 1,

Table 3: hyperparameters used

| Parameter | Neurogym RNN | split MNIST MLP |
|---|---|---|
| Weight update learning rate | 0.001 | 0.001 |
| Latent Update learning rates | 0.01 | 0.001 |
| Latent Update $\mathcal{L}_2$ regularization | 0.0001 | 0.1 |
| Latent size | 15 | 10 |
| Max latent updates | min(1000, $batchno$) | |
| $W^z$ sparsity | 0.5 | 0.8 |
| Loss function | MSE | MSE |

to prevent multiplying the RNN units inputs by values $< 1$ each time step leading to exponential decay. Additionally, since having negative gating values made little sense numerically or biologically, we rectified the values and set any negative values to 0. We found that having $\mathbf{W}^z$ drawn from a rectified Gaussian distribution with variance 1 had a similar performance to the baseline model with Bernoulli $\mathbf{W}^z$ (Fig A14). We show that sparsity matters and multiplying the Gaussian values with a Bernoulli improves performance further (Fig A15), presumably so a proportion of neurons and their weights are dedicated to some tasks. Additionally, the results reported used a fixed $\mathbf{W}^z$ and trained the other RNN parameters $\{W^r, W^{in}, W^{out}\}$. While this initially seemed to stabilize the learning problem to prevent the latent embedding meaning from changing rapidly, it the last version of the model, allowing all parameters to be trainable showed a reasonably similar performance, and fixing some of the weights is not a necessary assumption (Fig A16).

---

**Algorithm 1** Thalamus algorithm

---

Init $\mathbf{z}$ to a uniform value
**repeat**
    Sample $x^k, y^k$ from $D^k$ in sequential blocks
    $\hat{\mathbf{y}}^k = f_\theta(\mathbf{x}^k, \mathbf{z})$
    Accuracy = $\mathcal{L}(\hat{y}^k, y^k)$
    **if** Running mean accuracy - Accuracy $\geq 0.1$ **then**
        **while** remaining latent updates AND Accuracy $<$ Criterion **do**   ▷ latent updates ramp up
to a max of 1000
            $\Delta\mathbf{z} \leftarrow \alpha^z \frac{d\mathcal{L}}{d\mathbf{z}} + \mathcal{L}_2 reg$                     ▷ $\alpha^z$ learning rate was 0.01
            $\hat{\mathbf{y}}^k = f_\theta(\mathbf{x}^k, \mathbf{z})$
            Accuracy = $\mathcal{L}(\hat{\mathbf{y}}^k, \mathbf{y}^k)$
        **end while**
    **end if**
    **if** Running mean accuracy - Accuracy $\geq 0.1$ **then**
      $\Delta\mathbf{W} \leftarrow \alpha^W \frac{d\mathcal{L}}{d\mathbf{W}}$                ▷ Update weights only if accuracy not recovered
    **end if**
    **Update** Running mean accuracy
**until** Converged: not requiring any weight updates for the all tasks tested

---

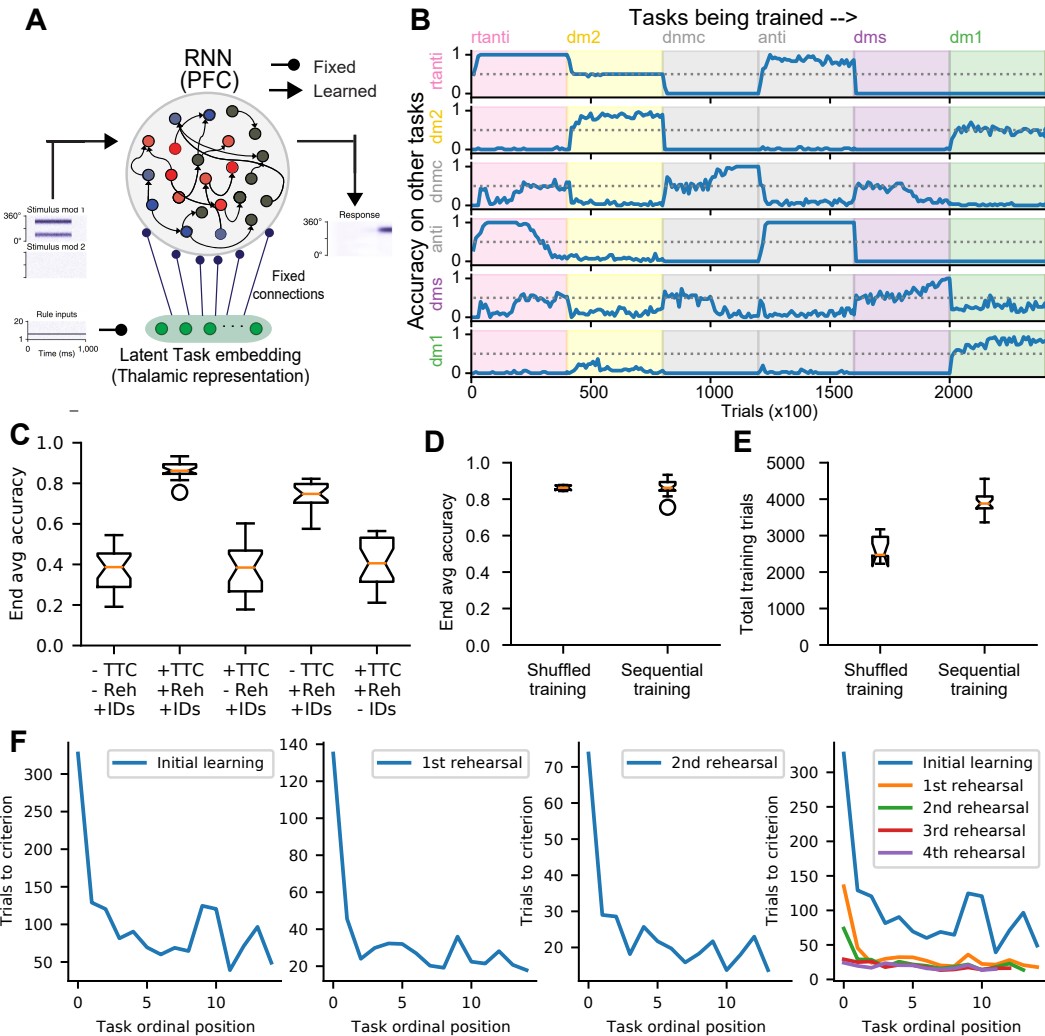

Figure 7: (A) Network schematic. (B) Accuracy on all tasks as the network learns them sequentially. (C) Accuracy at the end of the training session averaged across all 15 tasks, measured over 10 random orderings of the tasks. The x axis labels indicate which components were used in the training paradigm. TTC indicates training each task to criterion, Reh, indicates rehearsal of previous tasks after each new task, and IDs indicates whether task identity was input to the network. The combination of the three components was necessary for optimal accuracy. (D) For comparison, we train the network using shuffled data until it achieves similar accuracy and (E) compare the number of batches needed using shuffle training and sequential training. (F) Decreased batches to criterion as a function of the order of which task was learned in the initial learning and in subsequent rehearsals. The right subplot is the same lines overlaid to show progression.

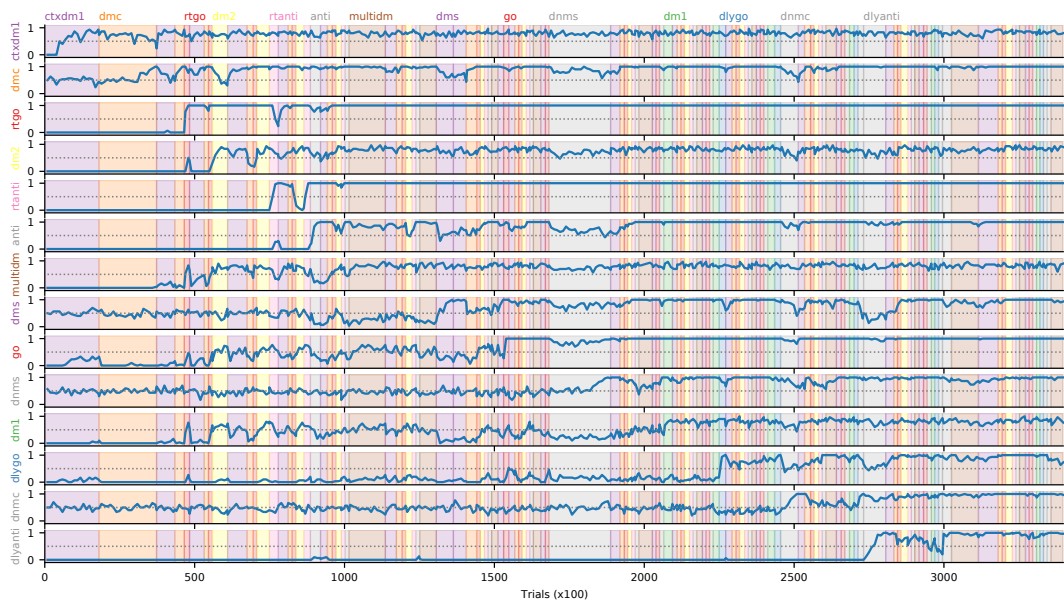

Figure 8: Training on all 15 tasks with rehearsal and training to criterion.

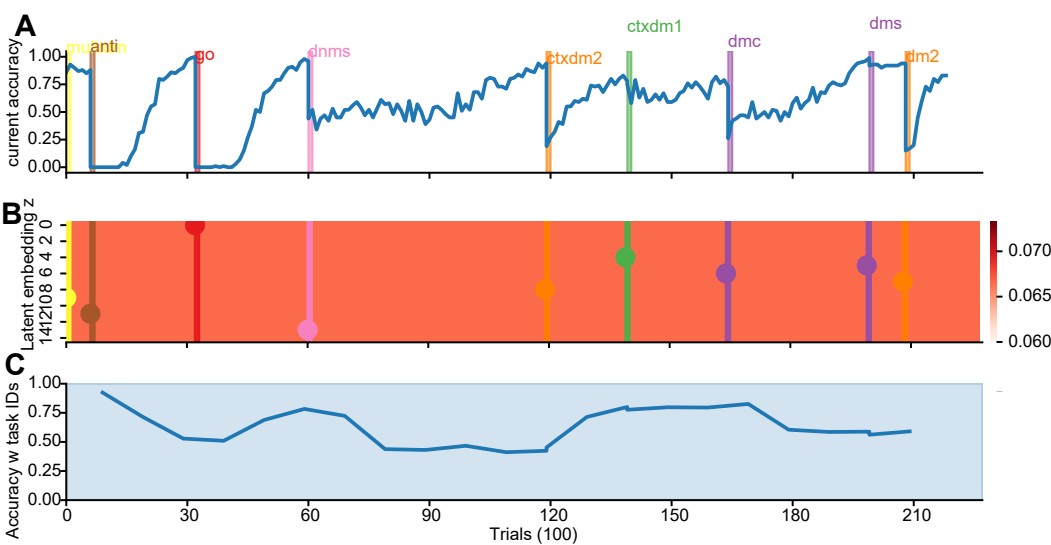

Figure 9: Pretrained network adapts rapidly to changing tasks. (A) solving a stream of tasks by rapidly fine-tuning the weights. (B) Latent task embedding z kept uninformative and uniform with the network no longer receiving task IDs as input. (C) the network begins to no longer respond to the pretraining task IDs, and accuracy on all tasks drops with progressive adaptation of the weights.

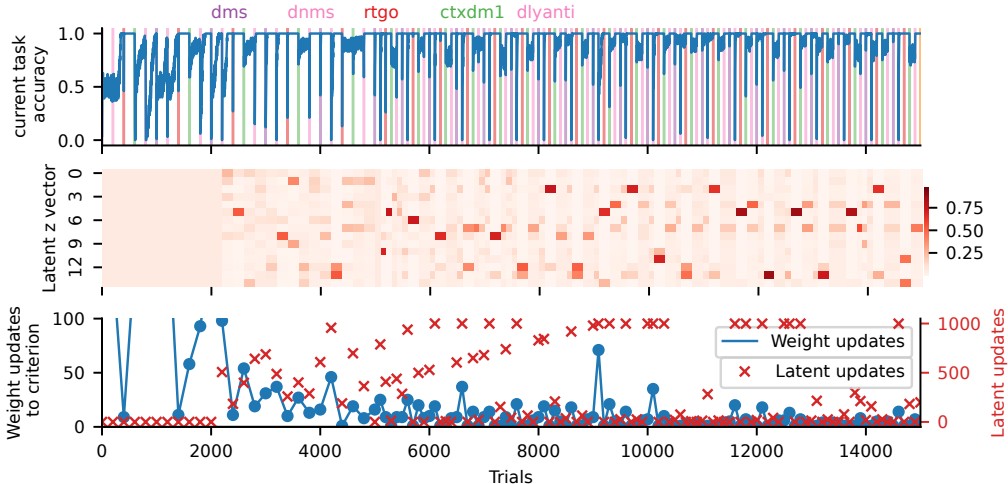

Figure 10: Newly initialized network learning unlabelled tasks using the Thalamus algorithm. Training begins with exposure to each task for 200 batches, and then for 100 for visualization purposes. Note that as opposed to Fig 2.A here we only show batches of new data points and omit latent update loops (A) Accuracy on current task. Vertical lines label transition between tasks, color coded to task identity. (B) latent vector representation. (C) Weight updates (left axis) and Latent updates (right) to reach accuracy criterion.

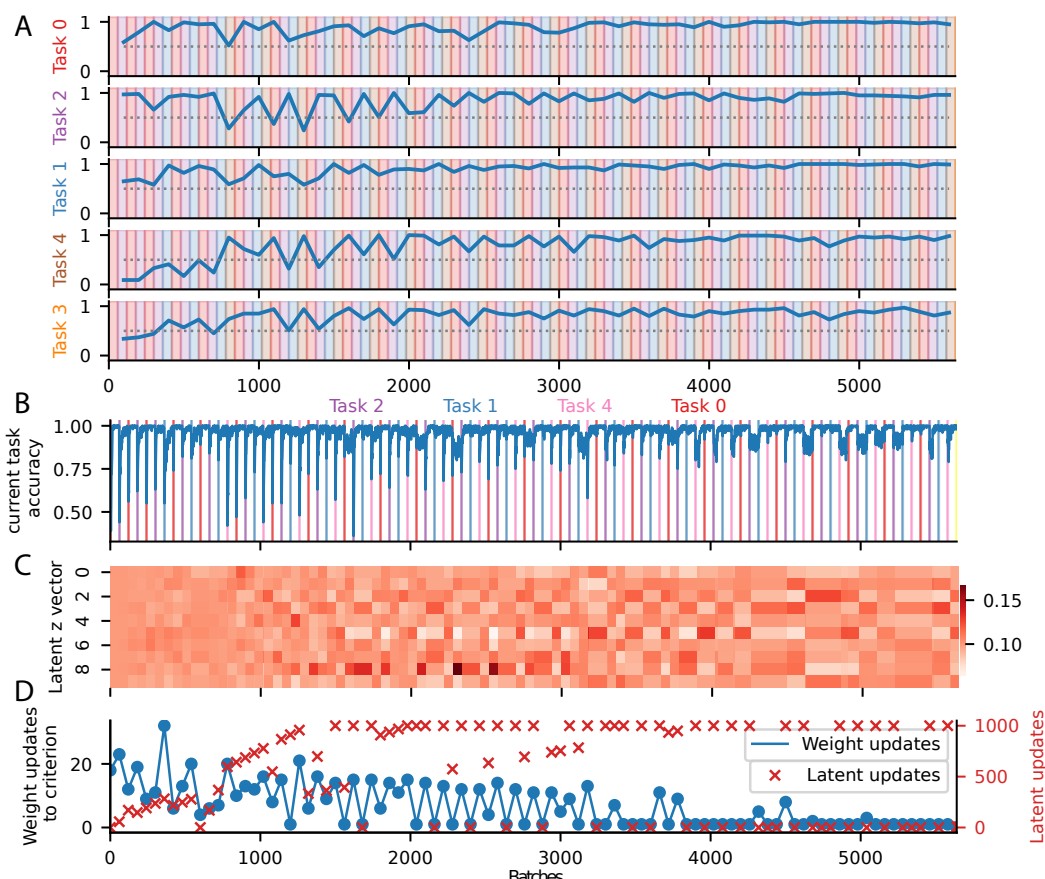

Figure 11: Example run of Thalamus on split MNIST tasks. We provide an example where the 5th task "task 3" was not seen through out the training sequence but performance on it increases after the model forms latent representations of the other tasks.

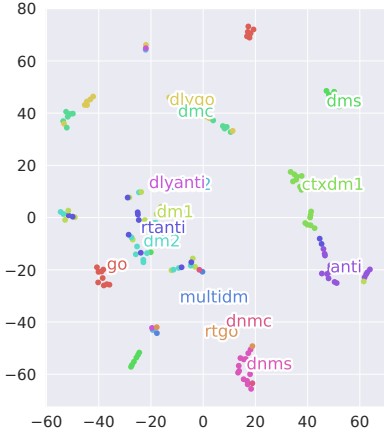

Figure 12: Diagram of T-SNE reduced latent embeddings learned from running Thalamus on 10 cognitive tasks. We froze the weights after 30 rehearsals to prevent semantic drift, and gathered data with no weight updates.

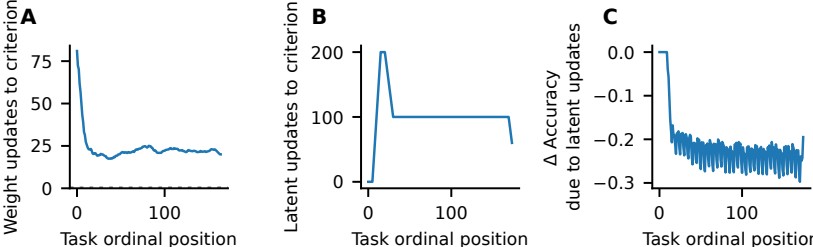

Figure 13: Showing the effects of updating weights and latent every batch instead of the accuracy criterion. A) The model continues to rely on weight updates and never approaches zero weight updates. B) Latent updates in each block simply reflect the length of the task block as the latent are updated every batch. C) The change in accuracy after the latent update indicates mainly a drop in accuracy after the latent updates. Results averaged over 20 random seeds.

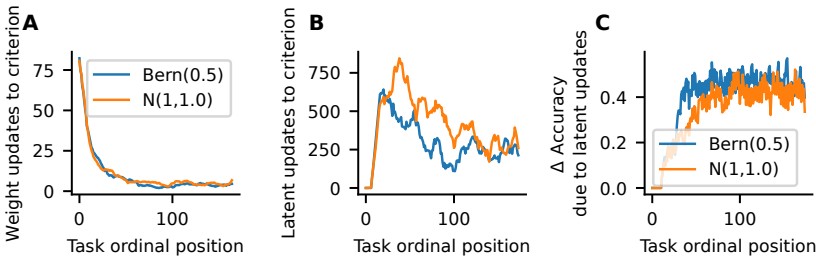

Figure 14: Thalamus performance metrics on 5 neurogym tasks. Results from 20 seeds. Comparing model with $\mathbf{W^z}$ drawn from a Bernoulli with (p = 0.5) vs. a rectified normal distribution with mean 1 and variance 1.

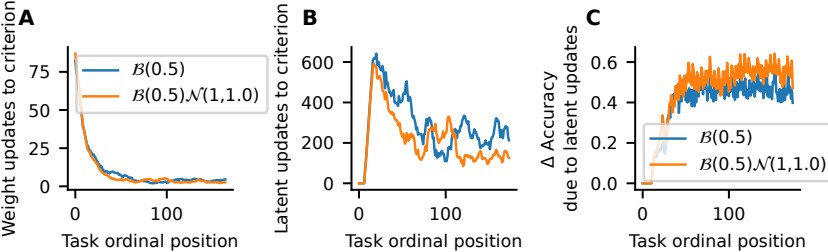

Figure 15: Same as Fig A14 but the rectified normal distribution with mean 1 and variance 1 further sparsified by multiplying with a Bernoulli (p=0.5).

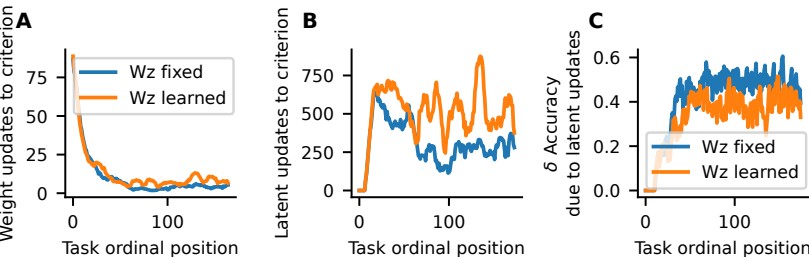

Figure 16: Thalamus performance metrics on 5 neurogym tasks. Results from 20 seeds. Comparing model 'Wz fixed', which is the baseline model used in the paper, to the model with $\mathbf{W^z}$ allowed to be trainable.

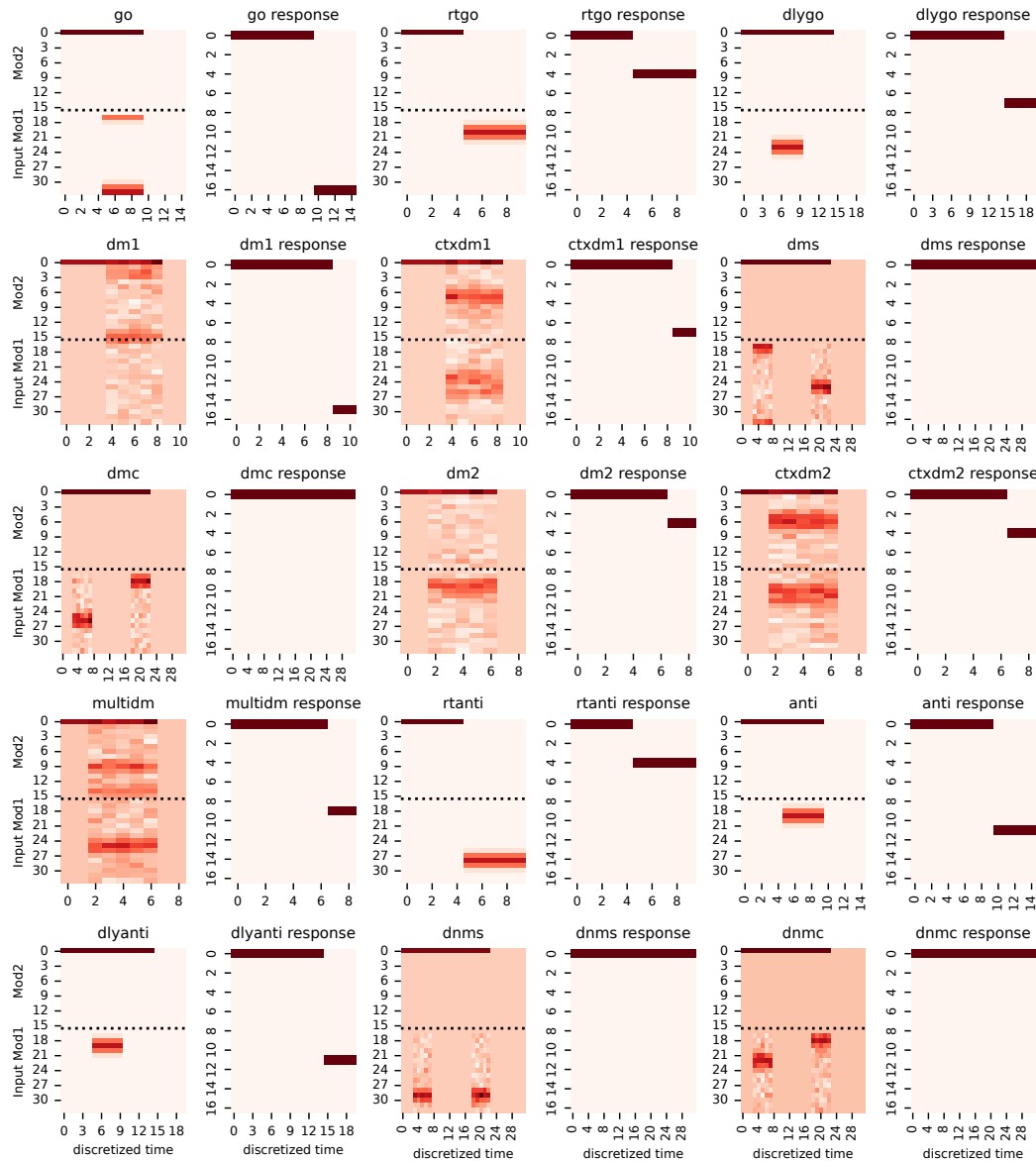

Figure 17: Visualizing the inputs and the ground truth responses for neurogym tasks. The first input dimension has a fixation signal, where agents are to withhold responses and attend to two sensory modalities (mod1, and 2). Inputs come as a pulse of various lengths at an angle in one modality. In 'Go' tasks, agent is to simply report the same angle as output in the same modality. 'Anti' tasks require reporting the opposite angle. Delay tasks have the input removed and response is delayed. In decision making tasks ('dm') Inputs are noisy across two modalities and the correct response to infer the peak of the noisy precept. Which modality to report on depends on context, and the type of task (for e.g. in ctxdm2, input in mod2 should be ignored). Delayed non-match to sample ('dnms'), and similar tasks require comparing to inputs held in memory and responding accordingly. See Yang et al. (2019) for further descriptions

