# OpenReview forum: "Thalamus: a brain-inspired algorithm for biologically-plausible continual learning and disentangled representations"
_ICLR.cc/2023/Conference — ICLR 2023 poster_

### Official Review · Reviewer_ugKD · 2022-10-23

**Confidence:** 3
**Correctness:** 3
**Technical Novelty And Significance:** 2
**Empirical Novelty And Significance:** 3
**Recommendation:** 6

**Clarity, Quality, Novelty And Reproducibility:**

The paper is clearly written and the logic is clear. However, the task should be introduced in a clearer way since many readers are not familiar with the cognitive tasks such as working memory and decision making.

**Strength And Weaknesses:**

The proposed model is simple and effective on the benchmark. The author claimed that the model shows generalization to novel tasks. However, with such simple network, I am wondering if this is a real generalization, i.e., how different the novel tasks is from those tasks the model has seen during training? Do they share the same underlying task distribution?

There are several minor questions here:

1, Why is the multiplicative effect consistent with the dis-inh. thalamocortical projections?

2, What is r(t) in Eq.5?

3, During training, why do you keep $W^z$ fixed?


**Summary Of The Paper:**

The authors proposed a RNN with latent embedding that uses optimization at inference time to generate internal contextual signals allowing the agent to parse its temporal experience into discrete events and organize learning about them. They showed that the model trained on tasks sequentially using weight updates with task identifiers provided, can be used to identify tasks dynamically by taking gradient steps in the latent space. Moreover, the model shows generalization to novel tasks and can discover temporal events at arbitary time-scale and does not require a pre-specified number of events or clusters.

**Summary Of The Review:**

The paper borrow some idea from how the brain carries out continual learning (especially relating to the Thalamus). But how the simple RNN model is related to the PFC and the Thalamus is not that clear. Even though the authors showed that the proposed model works well on the cognitive benchmark, it is still not clear to what extent the model can be applied to more realistic task (those tasks considered in the ML community).

---

> ### Author Response · Authors · 2022-11-15
> **Responses to reviewer ugKD**
>
> We thank the reviewer for their encouraging comments and questions.
>
> >*"Even though the authors showed that the proposed model works well on the cognitive benchmark, it is still not clear to what extent the model can be applied to more realistic task (those tasks considered in the ML community)."*
>
> We do apply the model to the standard split MNIST incremental learning task and compare to state-of-the-art models. This was not announced early nor clearly in the paper, as these experiments were in the very last section 3.4 “Comparison on external benchmark: split MNIST”. Since it is critical that these results are visible, we added several signposts at the beginning of Experiments section 3, and beginning and end of section 3.3.
>
> In the split MNIST task, the MNIST digits are split into 5 pairs, each pair made into a binary classification task (chosen arbitrarily). We quantify the end average test accuracy on the split MNIST task and compare with 5 continual learning methods including the most recently reported state-of-the-art model and Thalamus scores competitively (Table 2).
>
> In addition, we show that by learning the first four tasks, Thalamus can generalize to the fifth unseen task by taking latent updates with the weights frozen. The ability to generalize to the fifth unseen task does not emerge immediately after learning the first four, as likely there are many trivial ways to learn these tasks initially that do not generalize. Rather, it takes a few rehearsals with learning, forgetting, and then learning better features, until the model associates the latent embedding with computations that are general enough to be able to solve the unseen fifth task (Fig. 6).
> We also now better visualize the cognitive tasks in Fig A17, and we provide better description in the figure caption. Thanks for helping us make the connection to the ML community clearer.
>
> ---
> >*"… But how the simple RNN model is related to the PFC and the Thalamus is not that clear."*
>
> The reviewer raises a fair point, PFC is the seat of advanced cognitive abilities, how come it is modeled as an RNN still. While neuroscience elaborated the structural details of the PFC, the computational understanding still lags, and the most understood aspect remains the rich temporal dynamics that goes minutes into the past and the future. Influential neuroscience works have continued to view PFC as an RNN (examples, Mante et al., 2013, Wang et al., 2018, referenced below). Similarly, the thalamus is modeled as a non-recurrent set of neurons, reciprocally connected to the cortical model (examples, Logiaco et al., 2021, Kao et al., 2021)
>
> Regarding biological plausibility of our model, we use backprop to estimate gradients of the latent embedding, but recent ML works show that these gradients can be estimated by a second neural network (Marino et al., 2018, Greff et al. 2019). These learned credit assignment projections then would correspond to the corticothalamic projections while the projections from the latent embedding to the RNN would correspond to the thalamocortical projections.
>
> ---
> >*"1, Why is the multiplicative effect consistent with the dis-inh. thalamocortical projections?"*
>
> * After reflecting on this point, we agree, the multiplicative effects were observed experimentally as thalamic stimulation non-linearly amplifies PFC responses (Schmitt et al., 2018), but these effects are not evident from the dis-inhibitory structure alone. We adjusted the wording in the Model section 2.
>
> >*"2, What is r(t) in Eq.5?"*
>
> * It was never explicitly defined. It is the RNN activities after the relu non-linearity. We removed it entirely and used symbols already defined to describe.
>
> >*"3, During training, why do you keep Wz fixed?"*
>
> * Thanks for the thoughtful question. The reviewer intuition was correct to question the need for fixed weights. We tested the model with Wz fixed or trainable, and it minimally impacted performance. We added this now as Fig. 16. We now provide a detailed discussion in the appendix, of why the weights are fixed in the reported model and why that is not a necessary assumption (Section A.5 titled “Further examination of the latent embedding to RNN weights Wz”).
>
> ---
> Mante, V., Sussillo, D., Shenoy, K. V. & Newsome, W. T. Context-dependent computation by recurrent dynamics in prefrontal cortex. Nature 503, 78–84 (2013).
>
> Wang, J. X. et al. Prefrontal cortex as a meta-reinforcement learning system. Nat Neurosci 21, 860–868 (2018).
>
> Logiaco, L., Abbott, L. F. & Escola, S. Thalamic control of cortical dynamics in a model of flexible motor sequencing. Cell Reports 35 (2021).
>
> Kao, T.-C., Sadabadi, M. S. & Hennequin, G. Optimal anticipatory control as a theory of motor preparation: A thalamo-cortical circuit model. Neuron (2021).
>
> Greff, K. et al. Multi-Object Representation Learning with Iterative Variational Inference. ICML, 2019.
>
> Marino, J., Yue, Y. & Mandt, S. Iterative Amortized Inference. ICML 2018.

---

> > ### Comment · Reviewer_ugKD · 2022-12-02
> > **Thank you for your response**
> >
> > The authors addressed most of my questions here. I am happy to increase the score.

---

> ### Author Response · Authors · 2022-11-17
> **split MNIST helped?**
>
> Dear reviewer,
> As the discussion phase 1 ends in two days, we wonder if there is any feedback from examining the split MNIST results.
> We do acknowledge that quantifying generalization is hard, even hard to describe in words, so we are hopeful that with split MNIST there is enough familiarity that it addresses the concern the reviewer raised. It was helpful for us, as interpreting the cognitive task was hard, despite our familiarity with them.
>
> Please let us know if any further thoughts arise and we are happy to discuss or update the paper to clarify.
>
> Thanks again.

---

### Official Review · Reviewer_JbgB · 2022-10-24

**Confidence:** 3
**Correctness:** 3
**Technical Novelty And Significance:** 2
**Empirical Novelty And Significance:** Not applicable
**Recommendation:** 6

**Clarity, Quality, Novelty And Reproducibility:**

I am not a researcher in the field of continuous learning, but I am a researcher in brain-like computing, this writing and organization of paper makes me unable to judge its novelty and value. Generally, I feel the technical details and contributions of this paper are very vague, especially since the authors claim that their work benefits from brain mechanisms.

**Strength And Weaknesses:**

The paper is somewhat well-written. The results may show the effectiveness of proposed method.

**Summary Of The Paper:**

The paper proposes a brain-inspired algorithm for continual learning and disentangled representations. However, the contributions are generally not clear.

**Summary Of The Review:**

Weakness:
1、	The connection between brain mechanisms such as thalamic-cortex interaction is weak. The background description is very confusing. I could not get the clear message from the description. I feel that they purposely link their algorithm with a brain mechanism. Unfortunately, authors don't tell a complete and feasible story.
2、	The description for method is simple (e.g., the Section 2 contains very short paragraphs) but I can not see any connections between the model and the motivation mentioned above.

---

> ### Author Response · Authors · 2022-11-15
> **Response to reviewer JbgB**
>
> We are thrilled to have a brain-like computing researcher in this discussion, as implementation in neuromorphic hardware is one application we are excited about. But it does seem like our vision did not translate well and that is helpful to know. The updated manuscript now has a discussion of biological-plausibility (last paragraph in section 2) and also a statement about how we see this implemented in hardware (section 5). Here, would like to take the time and elaborate on how this algorithm would provide neuromorphic hardware with the ability to show adaptable behavior, solve new tasks, and ‘learn’ without needing backpropagation or expensive weight updates.
>
> Our model proposes iteratively learning a set of tasks, while alternating between updating the weights of the model and updating a latent embedding that gates the model computations. We show then that through this procedure, the model requires fewer and fewer weight updates to transition between unlabeled tasks and begins to rely on latent updates. Through rounds of learning, forgetting and relearning better features, it reaches a point where it can even solve an unseen task purely by updating the latent embedding.
>
> We are currently using backpropagation to get the gradients for the latent embedding values, which is not compatible with brain or hardware implementation. However, recent work in ML shows that these gradients can be effectively learned by another neural network (Marino et al., ICML, 2018, and Greff et al., ICML, 2019, references below). Once learned, these connections estimating latent gradients would correspond to the cortico-thalamic projections and open the path for the hardware implementation we are excited about. For this work, however, we use backpropagation to maintain generality, as amortizing the gradients imparts a level of domain specialization.
>
> The model would be implemented as two neural networks. One does the task computations and is gated by a latent embedding, and a second that compares the output to feedback from environment and produces gradients for the latent embedding. This two-neural-networks setup would give the hardware the ability to adapt its behavior (‘learn’) and generalize to new situations, without requiring backpropagation or updates in weights space that are difficult to implement and come at the cost of forgetting and unpredictable changes in behavior if the hardware was deployed. Very much how animals adapt by inferring change in the environment rather than updating all connections in their brains (Heald et al. 2021).
>
> Regarding biological plausibility, the architecture is compatible with the thalamocortical circuit, where the weights from the latent to the RNN correspond to the thalamocortical projections, and the credit assignment network corresponds to the corticothalamic projections. As such, the biological plausibility we are interested in is at an architectural level that corresponds to brain regions, and not at the level of the synapses that connects two neurons.
> We do not offer a theory of what learning rules the brain uses, and assume learning relies on the credit assignment mechanisms in the brain, though these are yet to be fully understood (Lillicrap et al., 2020). We follow recent work where models trained with backpropagation remain the field’s best models of brain function and neuronal responses (Richards et al., 2019).
>
> We hope this better clarifies the contribution of the model and the link to brain-like computing at a systems-neuroscience level. We are happy to address any further concerns or comments.
>
> ---
> Greff, K. et al. Multi-Object Representation Learning with Iterative Variational Inference. ICML, 2019
>
> Marino, J., Yue, Y. & Mandt, S. Iterative Amortized Inference. ICML, 2018.
>
> Heald, J. B., Lengyel, M. & Wolpert, D. M. Contextual inference underlies the learning of sensorimotor repertoires. Nature 600, 489–493 (2021).
>
> Lillicrap, T. P., Santoro, A., Marris, L., Akerman, C. J. & Hinton, G. Backpropagation and the brain. Nat Rev Neurosci 21, 335–346 (2020).
>
> Richards, B. A. et al. A deep learning framework for neuroscience. Nat Neurosci 22, 1761–1770 (2019).

---

> > ### Author Response · Authors · 2022-11-17
> > **Any further thoughts?**
> >
> > Dear reviewer,
> >
> > We hope that our earlier elaboration on biological plausibility and hardware implementation clarified the connection to brains. Please let us know of any other points that we need to clarify, and if any parts of the paper are still vague.
> >
> > We also have a specific question that would be of help. While we believe that the term 'biologically-plausible' has been used in a restrictive sense of late to only refer to Hebbian-like local learning rules, we do acknowledge that this is where the field is. We wonder if 'biologically-plausible' in our title biased your expectations of what the paper is, and if so, we are open to removing the term from the title of our paper to avoid future confusion.
> >
> > We hope for your timely response as the first discussion phase is ending in less than two days.
> >
> > Thanks again.

---

> > > ### Comment · Reviewer_JbgB · 2022-12-02
> > > **Thanks for your reply.**
> > >
> > > Thanks for your reply. I am happy to increase the score. However, I hope this paper could contribute to more introduction of related biological mechanisms adopt in this work. At present, the related reply is still quite weak. I suggest that the area chair may consider this point.

---

### Official Review · Reviewer_5VKC · 2022-10-25

**Confidence:** 2
**Correctness:** 3
**Technical Novelty And Significance:** 3
**Empirical Novelty And Significance:** 3
**Recommendation:** 8

**Clarity, Quality, Novelty And Reproducibility:**

Overall, the paper is clearly written and easy to follow. The following are some minor comments/questions I had:
- The symbols $t$, $\mathrm{t}$, $\mathbf{t}$ are all used;  it is a bit confusing as to which one corresponds to time and task
- The network output is described as a linear layer projection from the RNN neurons, but eq 5 also includes a relu activation function?
- What do the round colored markers in figure 2C correspond to?
- The loss function used to train the network is not described in the main paper or the appendix
- At the end of section 3.5, is the learning rate for SGD $10^{-4}$ ?

**Strength And Weaknesses:**

The proposed model is a simple RNN with projections from a latent embedding vector. But by training this model using a simple procedure of alternating RNN weight and latent embedding updates, we obtain a network that is capable of learning tasks in the absence of any explicit task labels. These results are quite interesting, and as the authors mention in the conclusion there are a lot of interesting future directions (e.g., using a latent space that allows for compositionality over computational primitives).

The following are some comments/questions I had:
-  Is there a specific reason why the weights from the latent embedding are drawn from a Bernoulli distribution?
- Figure 3 shows that the latent embeddings cluster according to task id with the pretrained network. I suppose the same could be expected in the thalamus network trained from scratch?
- What is the loss function used to train the networks?
- I wonder if there is a better criterion for switching to latent updates than just a drop in accuracy. This requires the network to always know how well it is performing at the task?
- Would simultaneously optimizing both the RNN weights and the latent embedding vector do worse than the alternating updates?

**Summary Of The Paper:**

This paper presents a model for continual learning that is inspired by thalamocortical networks in the brain. Neuroscience studies have revealed that (i) the prefrontal cortex (PFC) shows representation of relevant task variables, and (ii) the thalamus shows representations of the task being performed. Further, is thought that the connections from the thalamus to the PFC gate the computations performed by the PFC by selecting the relevant representations.

In the proposed model, a RNN and a latent embedding vector serve as abstractions of the PFC and the thalamus, respectively. The projections from the latent embedding to the neurons in the RNN correspond to thalamocortical projections. The model is trained on a sequence of tasks with alternating RNN weight and latent embedding updates, resulting in a network capable of parsing a sequence of inputs into the appropriate contexts.

Finally, the performance of the model was evaluated on cognitive tasks from (Yang et al., 2019) and on the split MNIST task.

**Summary Of The Review:**

The paper is well written and the main concepts are clearly conveyed. The experimental results also highlight the potential of this approach. Overall, the proposed model is promising and could potentially lead to more interesting avenues of future work.

---

> ### Author Response · Authors · 2022-11-15
> **Response to reviewer 5VKC**
>
> >*"Overall, the proposed model is promising and could potentially lead to more interesting avenues of future work."*
>
> * We thank the reviewer for their encouraging remarks, for especially for recognizing the future potential of this work.
>
> The reviewer asks a set of thoughtful questions, and even provides ideas for future direction. We tested these ideas, and they did indeed lead to interesting results that improved our understanding of the model. The reviewer self-rated their confidence humbly at 2.
>
> ---
> >*"Is there a specific reason why the weights from the latent embedding are drawn from a Bernoulli distribution?"*
>
> * Thank you for this comment. The issue with weights from a Gaussian distribution was small values multiplying the RNN activities every time step leading to exponential decay of network activity. But this comment inspired us to now run simulations with weights from a Gaussian with mean 1, and it indeed showed effective performance. These new results are now added in Fig. 14. We rectified the Gaussian weights, as multiplying the RNN inputs with negative values seemed implausible. So there is a proportion of zero values. To confirm that the portion of zeros is the critical feature of these weights, we multiplied the Gaussian with a Bernoulli to increase sparsity and indeed saw a performance boost. We updated the model description section 2 paragraph 4 to indicate either a Bernoulli or rectified Gaussian work, and we added a new appendix section to discuss these findings. Thanks for enhancing the generality of the model.
>
> ---
> >*"Figure 3 shows that the latent embeddings cluster according to task id with the pretrained network. I suppose the same could be expected in the thalamus network trained from scratch?"*
>
> * We have not tried that previously due to semantic drift reported in Fig. 5. We ran 10 cognitive tasks and then froze the weights to prevent drift, and then collected data for t-SNE. We now added this as Figure 12 and indeed we see some functional clustering, but also some unexpected phenomena with some tasks having two clusters and some tasks sharing one or several clusters. Seems like a complex surface for future studies.
>
> >*"What is the loss function used to train the networks?"*
>
> * We apologize for this omission. We used the MSE loss. Added to section 2 and table 3.
>
> ---
> >*"I wonder if there is a better criterion for switching to latent updates than just a drop in accuracy. This requires the network to always know how well it is performing at the task?"*
>
> * This is a great suggestion. We are looking at other structures, like a world model for example, and surprising events, that are not predicted by the world model, would trigger latent updates. Another approach would be monitoring the statistics of the loss function, so an absolute scale is not needed, but rather significant deviations for most recent loss value is sufficient. This is ongoing work, if either of these works well before the first discussion phase we might update the paper.
>
> ---
> >*"Would simultaneously optimizing both the RNN weights and the latent embedding vector do worse than the alternating updates?"*
>
> * Yes, unfortunately, it would perform worse. We ran this simulation where we update the weights and then the latents every batch of data. It was helpful to see in what ways the model broke down. These results are added as Fig. 13 in the updated paper. We saw that latent updates now on average decrease accuracy. The weight updates are trying to associate the computation with a moving latent target. Intuitively, once the decision is made to update weights, it is important that the latent stays somewhat stable, and if the latent solves the task, then it is important to avoid making any weight updates to avoid forgetting.
>
> ---
>
> >*"Overall, the paper is clearly written and easy to follow. The following are some minor comments/questions I had:"*
>
> * Thank you for the set of comments that improved the readability of our paper. We addressed the points raised as follows.
>
> >*"The symbols t, t, **t** are all used; it is a bit confusing as to which one corresponds to time and task"*
>
> * Thanks for noting this. We changed the task identifier to k
>
> >*"The network output is described as a linear layer projection from the RNN neurons, but eq 5 also includes a relu activation function?"*
>
> * We removed the description ‘linear’; the equation is correct. Thanks.
>
> >*"What do the round colored markers in figure 2C correspond to?"*
>
> * They correspond to the one-hot encoded vector we supplied to the network for learning. This was never explained, and we now updated the figure caption.
>
> >*"At the end of section 3.5, is the learning rate for SGD 10^−4^ ?"*
>
> * Yes. Fixed. Thank you.

---

> > ### Comment · Reviewer_5VKC · 2022-12-02
> > **Thank you for your response**
> >
> > I appreciate your thoughtful response. All of my questions have been addressed. And thank you for also performing new experiments based on my questions/comments. I am happy to update the score to 8 as well.

---

### Official Review · Reviewer_W9vo · 2022-10-25

**Confidence:** 3
**Correctness:** 4
**Technical Novelty And Significance:** 3
**Empirical Novelty And Significance:** 3
**Recommendation:** 8

**Clarity, Quality, Novelty And Reproducibility:**

### Clarity

The paper is well-written and easy to follow. Some additional discussion could improve the presentation/place it better in the context of biologically plausible learning.

Fig. 4C: is the average done over random seeds? How many, and what’s the average latent update count? Ideally it would be plotted on a separate graph (in the appendix?).

### Quality

The technical side of the paper seems solid. The experiments are done on a many tasks and are compared with other continual learning algorithms.

Question about prompting with a training batch (Tab. 2): are there any updates in the weights during prompting, or only in the latents? I understand the need to do it, but **I think the fair way to compare Thalamus to other algorithms would be by freezing the RNN weights completely during evaluation** (I'm not sure if it's done or not).

**Minor issues:**

Overall, the graph captions/labels could be made more clear and aligned among subplots.

Eq.5: (t) should not be a subscript (to be consistent with the rest). Maybe replacing relu with \phi (or replacing the \phi earlier with relu) would improve readability.

End of page 5: random square in the middle of the page

Fig. 4A: labels on top are not readable. I think removing them completely, and in addition only plotting the final accuracy for each task would massively improve the plot. The running accuracy plot (Fig. 4A) could be then moved to the appendix as learning curves don't tell us much here.

Fig. 11 is cut short at the bottom.

### Novelty

The work is, to my knowledge, novel. The related papers on the computational role of thalamus are mentioned, but the authors should have a look at the following paper:

Laureline Logiaco, L.F. Abbott, Sean Escola,
Thalamic control of cortical dynamics in a model of flexible motor sequencing, 2021

It introduces a different learning algorithm but is still very related to the current work.

### Reproducibility

The code is not provided, although Sec. 3 implies it will be. The authors can attach it as a zip to the submission.

**UPDATE**: the code is now provided.


**Strength And Weaknesses:**

### Strengths

A clear algorithm for continual learning that involves a thalamus-like part and offloads some of the weight updates in the main network to that part.

Good experimental performance.

A clear way to learn from an unlabeled stream of tasks (by triggering learning of z with large accuracy deviations) that also enforces stability in the RNN weights (as z is updated first in those cases).

### Weaknesses

As the goal of the paper is biological plausibility, it needs a discussion on how Thalamus could be implemented in the brain. As I can see it, the main issue would be credit assignment to the latents, requiring backpropagation through the RNN.

Presentation is lacking in parts (mostly the plots) but can be easily improved.

No code attached.

Comparison with other methods could be improved (see in bold below).



**Summary Of The Paper:**

The work couples an RNN with a latent space representation that can infer the current task in a continual learning setup. The latent space is inspired by the role of thalamus, and enables the RNN to adapt to new tasks by changing the latent representation rather than the internal dynamics of the RNN.

**Summary Of The Review:**

Good paper, but several aspects (presentation, discussion on plausibility, maybe better comparison with other methods, and adding code) could be improved. I put the current score to 6 but I can raise it if my concerns are addressed.

**UPDATE**: updated the score from 6 to 8 as all of my concerns have been addressed.

---

> ### Author Response · Authors · 2022-11-15
> **Response to reviewer W9vo**
>
> >*"A clear algorithm for continual learning.. Good experimental performance.. A clear way to learn from an unlabeled stream of tasks."*
>
> * We thank the reviewer for the encouraging comments.
>
> >*“As the goal of the paper is biological plausibility, it needs a discussion on how Thalamus could be implemented in the brain. As I can see it, the main issue would be credit assignment to the latents, requiring backpropagation through the RNN.”*
> * We absolutely agree that the paper is now strengthened by a discussion of brain implementation, now in section 2, last paragraph in the updated paper. We discuss further here.
> * The backpropagated gradients assigning credit to latents can be learned. Our model alternates between updating the parameters of the model and updating the latent embedding contextualizing the model responses. We use backpropagation to compute gradients for the latents, but recent ML works show that credit assignment to the latents can be learned by a second neural network (Marino et al., 2018, and Greff et al., 2019, referenced below).
> * The model would be implemented as two neural networks. One does the task computations and is gated by a latent embedding, and a second that compares the output to feedback from environment and produces gradients for the latent embedding. These learned credit assignment projections then would correspond to the corticothalamic projections while the weights from the latent embedding to the RNN would correspond to the thalamocortical projections.
> * In addition, once the credit assigning corticothalamic projects are learned, the structure can now show ability to adapt its behavior with no need for backpropagation or any parameters updates. The circuit now adapts by updating in latent space, rather than re-learning parameters, which matches modern formulations of adaptive behavior in neuroscience (Heald et al., 2021).
> * Finally, for this work, we use backpropagation to maintain generality, as amortizing the gradients imparts a level of domain specialization.
>
> Greff, K. et al. Multi-Object Representation Learning with Iterative Variational Inference. ICML, 2019
>
> Marino, J., Yue, Y. & Mandt, S. Iterative Amortized Inference. ICML, 2018.
>
> Heald, J. B., Lengyel, M. & Wolpert, D. M. Contextual inference underlies the learning of sensorimotor repertoires. Nature 600, 489–493 (2021).
>
> ---
> >*"I think the fair way to compare Thalamus to other algorithms would be by freezing the RNN weights completely during evaluation (I'm not sure if it's done or not)."*
> * The reviewer brings up a critical point regarding comparison to other algorithms. We agree, and this was indeed the approach we took. We now make this explicitly clear in the text that during evaluation, RNN weights were frozen, and only the latent embedding was allowed to change. We now mention this explicitly in the evaluation on the cognitive tasks and split MNIST in several locations. Thanks for pointing out this critical point.
> ---
> >*"No code attached."*
> * Apologies for the earlier omission. We now provide an anonymized repository at link posted as a comment to this forum accessible the reviewers and area chair.
> ---
> The reviewer made several comments that significantly improved the clarity of the paper. We addressed them in the updated version, here we address two of the points.
>
> >*"Fig. 4A: labels on top are not readable."*
> * We agree that the figure is overly complex and difficult to interpret. We reorganized the figure and removed unnecessary elements, and in fact moved it all to the appendix and replaced it with figure 5. We believe these data are a sufficient description of model behavior with an added subplot showing improvements in accuracy attributable to latent updates.
>
> >*"Fig. 4C: is the average done over random seeds? How many, and what’s the average latent update count? Ideally it would be plotted on a separate graph (in the appendix?)."*
> * We agree, this information is helpful, and this is exactly what we had plotted in figure 5. We now elaborated figure 5 and replaced figure 4 with it as described above. In the new Fig. 4, subplot A shows how reliance on weight updates decreases over time, and subplot B shows continued reliance on latent updates to adapt to task distribution changes. These are averaged over 20 random seeds.
>
> >*“The work is, to my knowledge, novel.. but the authors should have a look at the following paper: Laureline Logiaco, et al. .. 2021”*
> * Thank you for pointing out this paper. We now cite this paper, and even reached out and met the author! Much obliged.
> ---
> >*“Good paper, but several aspects (presentation, discussion on plausibility, maybe better comparison with other methods, and adding code) could be improved. I put the current score to 6 but I can raise it if my concerns are addressed.”*
> * We thank the reviewer for the encouraging remarks, we are open to address any further concerns, if any remain.

---

> > ### Comment · Reviewer_W9vo · 2022-11-15
> > **Good response, upped the score to 8**
> >
> > Thank you for a detailed response!
> >
> > All of my comments have been addressed, so I'm happy to update the score to 8.
> >
> > Regarding biological plausibility, I appreciate the discussion in the updated paper and in this comment. A second network that learns credit assignment makes sense and could work (at least there are papers like Akrout, Mohamed, et al. "Deep learning without weight transport." (2019) that do sort of that). It looks like there's some space left for the 9 page limit, so I think expanding the discussion at the end of section 2 with your comments here and maybe a more concrete architecture would be very helpful for readers. This can be done in the final version.

---

> > > ### Author Response · Authors · 2022-11-17
> > > **Thank you!**
> > >
> > > Dear Reviewer,
> > >
> > > Thank you again for the additional comments that continue to improve the paper. We agree again, the comments here about biological plausibility would enrich the paper and we will expand that section with the same comments we made here.
> > >
> > > Thanks for pointing us to that interesting literature. They do seem to learn the weights for the backward pass, and it is quite interesting. Possibly even without losing generality as they approximate backpropagated gradients.
> > >
> > > Thanks again, and please let us know of any further thoughts.

---

### Author Response · Authors · 2022-11-15
**Thank you for your feedback! Changes in the updated paper.**

We thank the reviewers for their thoughtful comments that not only improved the paper but also clarified connection to brain circuits and enhanced its generality. We were glad the reviewers found the paper “well-written” (Reviewers W9vo, 5VKC, and ugKD), the model “simple and effective” (Reviewer ugKD), and the results “quite interesting” and “novel” (Reviewers 5VKC and W9vo). The reviewers recognized not only the current contribution of the model “A clear way to learn from an unlabeled stream of tasks” (Reviewer W9vo) but also its potential with a “lot of interesting future directions” (Reviewer 5VKC). In the words of reviewer 5VKC, “The experimental results highlight the potential of this approach. Overall, the proposed model is promising and could potentially lead to more interesting avenues of future work.”

Following the insightful comments, we made several improvements in our updated paper, summarized below:
* The paper is now strengthened with a discussion of biological plausibility based on comments from reviewers (W9vo, 5VKC, JbgB), right after we introduce the model, end of section 2, last paragraph.
* An anonymized code repository will be posted to this forum implementing the algorithm and the analyses in the paper.
* Figure 4 was simplified, as suggested, moved to appendix, and replaced by adding a panel to Fig 5 (now Fig 4) (Reviewer W9vo).
* Added a statement about implementation in neural hardware (Reviewer JbgB) (section 5 sentence 5).
* Added Fig 16 to show that our assumption of fixed Wz weights from latent embedding to RNN were indeed unnecessary (Reviewer ugKD)
* Added Fig 12 to showing T-SNE plot of the latent embeddings Thalamus generated for tasks (Reviewer 5VKC).
* Added Fig 13 to show the necessity of the accuracy-based rule to engage latent updates was indeed a necessary component of the algorithm (Reviewer 5VKC).
* Added Figs 14 and 15 to show that our assumption of restricting Wz to a Bernoulli distribution also was an unnecessary assumption (Reviewer 5VKC).
* Added several sentences to clearly indicate that we go beyond the cognitive tasks and test on a standardized benchmark using split MNIST incremental learning task ((section 3 paragraph 1, section 3.3 paragraph 2) (Reviewer ugKD).

---

### Decision · Program_Chairs · 2023-01-20

**Decision:**

Accept: poster

**Justification For Why Not Higher Score:**

While there is general consensus that the paper should be accepted some of the claims regarding neuroscience were not adequately backed up.

**Justification For Why Not Lower Score:**

There is general consensus that this should be accepted.

**Metareview: Summary, Strengths And Weaknesses:**

This paper presents a model for continual learning that is inspired by thalamocortical networks in the brain. There is consensus that the paper should be accepted on the ground that the approach is interesting and it opens up novel avenues for future work and that the experimental evaluation is solid. There was two reviewers with more of a neuroscience background who found that the claims regarding the biological plausibility were overblown and I would encourage the author to tone these down.

**Note From Pc:**

if the above contains the word "oral" or "spotlight" please see: "oral" presentation means -> notable-top-5% and "spotlight" means -> notable-top-25%. As stated in our emails, we are disassociating presentation type from AC recommendations